# OpenReview forum: "To Think or Not to Think: Exploring the Unthinking Vulnerability in Large Reasoning Models"
_ICLR.cc/2026/Conference — Submitted to ICLR 2026_

### Official Review · Reviewer_WiJe · 2025-10-24

**Soundness:** 2
**Presentation:** 3
**Contribution:** 2
**Rating:** 2
**Confidence:** 4

**Summary:**

This paper investigates the vulnerability of large reasoning models (LRMs) to “break of thought” attacks, where manipulating reasoning delimiters can cause the model to skip reasoning and produce direct answers. The authors design two types of Break of Thought (BoT) attacks: training-time backdoor injections during SFT or DPO, and inference-time adversarial suffixes, including both white-box and black-box variants. Experiments on MATH-500 and AIME-2024 show that these attacks achieve high success rates, reducing reasoning length and accuracy. To mitigate this issue, the paper proposes a “Monitor of Thought” (MoT) framework that detects when reasoning should stop or be shortened, improving computational efficiency and safety alignment. Overall, the work highlights the structural fragility of explicit reasoning processes in LRMs and calls for more robust reasoning architectures.

**Strengths:**

1. The figures are clean, well-labeled, and paired with informative captions.
2. The mathematical notation and formula presentations are clear and consistent.

**Weaknesses:**

1. The structure of this paper feels disjointed. Although Section 4 introduces the BoT unthinking attack and Section 5 presents the MoT defense, there is no subsequent connection between the two. Section 5 appears entirely independent of Section 4. Why didn’t the authors test how effectively MoT can defend against BoT attacks?

2. The paper mainly combines several existing techniques under the reasoning setting. For example, BoT is essentially a mix of backdoor and GCG attacks, while MoT resembles speculative decoding. Including so many technical tricks in one paper makes the overall contribution appear less innovative.

3. Many important baselines are missing. For BoT, similar attack methods already exist, yet the authors neither explain the differences from prior work nor include them as baselines [1-3]. Likewise, for MoT, numerous speculative decoding methods exist, but the authors fail to compare against or justify their novelty relative to those baselines [4-5].

4. The reference for line 370 is missing.

5. The reasoning tasks considered are too limited. For example, Table 1 only evaluates mathematical reasoning. How does the proposed method perform on other benchmarks?

> [1] Xiang Z, Jiang F, Xiong Z, et al. Badchain: Backdoor chain-of-thought prompting for large language models[J]. arXiv preprint arXiv:2401.12242, 2024.
>
> [2] Xu R, Qi Z, Xu W. Preemptive answer" attacks" on chain-of-thought reasoning[J]. arXiv preprint arXiv:2405.20902, 2024.
>
> [3] Guo Z, Tourani R. Darkmind: Latent chain-of-thought backdoor in customized llms[J]. arXiv preprint arXiv:2501.18617, 2025.
>
> [4] Miao X, Oliaro G, Zhang Z, et al. Specinfer: Accelerating large language model serving with tree-based speculative inference and verification[C]//Proceedings of the 29th ACM International Conference on Architectural Support for Programming Languages and Operating Systems, Volume 3. 2024: 932-949.
>
> [5] Zeng X, Shang Y, Chen J, et al. Root defence strategies: Ensuring safety of llm at the decoding level[J]. arXiv preprint arXiv:2410.06809, 2024.

**Questions:**

See weakness.

---

> ### Author Response · Authors · 2025-11-26
> **Author Response (Part 1)**
>
> Dear Reviewer WiJe,
>
> Thank you for your thoughtful review and for acknowledging the clarity of our figures and mathematical presentations. We appreciate your constructive feedback, which has helped us better articulate our contributions. We address your concerns below:
>
> ---
>
> ## **Response to Weaknesses**
>
> ### **W1: Why doesn't MoT defend against BoT attacks?**
>
> We appreciate this important question, as it touches the core design philosophy of our work. **The key clarification is that BoT and MoT are not intended as an attack-defense pair, but rather represent two different applications of the same underlying vulnerability** – one malicious (BoT by adversaries) and one beneficial (MoT by model owners).
>
> - **Conceptual distinction**: BoT demonstrates how adversaries can exploit the unthinking vulnerability to compromise model reliability through backdoor injection or adversarial suffixes. In contrast, MoT shows how model owners can intentionally leverage this same delimiter-manipulation mechanism to improve efficiency and safety. They operate in fundamentally different threat models: BoT assumes an adversary who poisons training data or crafts adversarial inputs, while MoT assumes a legitimate model owner who controls the deployment pipeline.
>
> - **Why they address different problems**: Testing "whether MoT defends against BoT" would be analogous to asking whether a legitimate API feature can defend against a backdoor attack – they serve orthogonal purposes. MoT is designed to terminate redundant or unsafe reasoning during normal operation, not to detect or mitigate adversarial manipulation of delimiters. The defense against BoT is the "thinking recovery alignment" strategy we propose in Section 4.4 and Appendix E.7.
>
> - We acknowledge that **exploring the interaction between BoT and MoT could provide additional insights**. If MoT's external monitor could detect abnormal reasoning patterns caused by BoT triggers, this would be an interesting direction. However, this was beyond the scope of our current work, which focuses on systematically characterizing the vulnerability itself and demonstrating both its risks and opportunities.
>
> ---
> ### **W2: Are BoT and MoT merely combinations of existing techniques?**
>
> We respectfully disagree with this characterization. **Our core contribution is not technical novelty in isolation, but rather the systematic discovery and characterization of a fundamental vulnerability in LRM architectures**, along with comprehensive demonstrations of how it can be exploited and utilized.
>
> - **On BoT's novelty**: While BoT employs backdoor training and adversarial optimization as building blocks, **the attack paradigm itself is novel**. Traditional backdoor attacks aim to corrupt the *content* of reasoning traces or inject malicious reasoning steps. In contrast, BoT targets the *structural mechanism* of reasoning by manipulating thought delimiters to completely bypass the thinking process. This is fundamentally different: we're not changing what the model thinks, but whether it thinks at all. Similarly, GCG has been used for jailbreaking to elicit harmful content, but we repurpose it to suppress reasoning – a distinct objective with different implications. **The novelty lies in recognizing that delimiter tokens constitute a single point of failure** and demonstrating multiple pathways to exploit this (template modification, backdoor, adversarial suffixes, transfer attacks).
>
> - **On MoT's novelty**: MoT is distinct from speculative decoding. Speculative decoding focuses on *accelerating inference* through parallelization and draft-verify mechanisms, treating the task as a computational efficiency problem. MoT, in contrast, performs **meta-cognitive evaluation** – assessing whether the *content quality* of reasoning is sufficient or risky, not just whether tokens can be predicted faster. Our external monitor evaluates semantic properties (e.g., "Is the reasoning becoming circular?" or "Does this trajectory head toward unsafe outputs?") rather than token-level probabilities. Appendix F.7 shows that even heuristic rules can serve as monitors, demonstrating this is fundamentally a content-awareness problem. Furthermore, while speculative decoding requires architectural changes, **MoT is a plug-and-play framework** that works with any LRM using thought delimiters.
>
> - **The broader contribution**: Beyond individual techniques, we provide the first systematic study of the unthinking vulnerability across 8 LRMs, multiple attack vectors (training-based, training-free, white-box, black-box), adaptive defenses (Appendix E.5), and diverse applications (efficiency and safety). This comprehensive characterization has significant implications for LRM architecture design.

---

> ### Author Response · Authors · 2025-11-26
> **Author Response (Part 2)**
>
> ---
>
> ### **W3: Missing baselines**
>
> Thank you for pointing out the need for more comprehensive baseline comparisons. We acknowledge that our paper would benefit from more explicit comparisons to related work, and we appreciate you providing specific references [1-5]. We explain below how our work relates to these baselines and clarify the key differences.
>
> - **For BoT**: The suggested baselines [1-3] address related but distinct problems. BadChain [1] injects malicious reasoning steps but does not suppress thinking entirely. Preemptive Answer [2] explores how LLMs can be guided toward direct answers but does not systematically study backdoor or adversarial attacks on reasoning. DarkMind [3] focuses on latent backdoors in customized LLMs, whereas we target the explicit delimiter mechanism. **We acknowledge these are relevant related works and have added comparisons in Appendix E.2**, showing that straightforward adaptations of these methods (e.g., BadChain_unthink) achieve near-zero ASR, while our BoT consistently achieves >95% ASR. This validates that our approach is not a trivial extension.
>
> - **For MoT**: We agree that explicit comparisons with TokenSkip [1] and ThoughtMani [2]  would strengthen the paper. TokenSkip focuses on controllable CoT compression through token-level skipping, while ThoughtMani explores external thought manipulation – both are efficiency-focused. **MoT differs in providing a unified framework for both efficiency and safety** through external monitoring, which these works do not address. We will add detailed comparisons in the revised manuscript. Regarding speculative decoding methods [4-5], as explained above, they solve a different problem (computational acceleration vs. content-aware reasoning control). We will clarify these distinctions in the related work section.
>
> ---
>
> ### **W4: Limited reasoning tasks in main results**
>
> We appreciate this observation and want to clarify that **our evaluation actually covers diverse reasoning domains beyond mathematical tasks**, though space constraints in the main paper led us to focus on MATH-500 and AIME 2024 for consistency.
>
> - **Why we focused on mathematical reasoning in Table 1**: Mathematical reasoning tasks are particularly well-suited for demonstrating the unthinking vulnerability because (1) they require multi-step reasoning where the thinking process is critical for correctness, making the impact of bypassing reasoning clearly measurable through accuracy metrics, and (2) they are widely recognized benchmarks in the LRM community, allowing direct comparison with baseline model performance. This choice enables readers to clearly see the causal relationship between suppressing reasoning and performance degradation.
>
> - **Comprehensive evaluation across diverse domains**: However, we have extensively evaluated BoT attacks across multiple reasoning categories. **In Appendix E.6 (Table 12)**, we present detailed results showing that the unthinking vulnerability is not limited to mathematical reasoning:
>
>     - **Multi-hop reasoning** (BBH logical deduction task): ASR = 96.0%, RPC = -40.7%, RTC = -80.3%
>     - **Commonsense reasoning** (HellaSwag, 1000 samples): ASR = 97.5%, RPC = -31%, RTC = -95.2%
>     - **Code generation** (LiveCodeBench, 2024-08 to 2025-02): ASR = 97.3%, RPC = -60.1%, RTC = -98.4%
>
> - These results demonstrate that **the unthinking vulnerability is a fundamental architectural issue that affects reasoning across diverse domains**, not an artifact specific to mathematical tasks. The consistently high ASR (>96%) across all domains confirms that delimiter manipulation reliably suppresses thinking regardless of task type, while the substantial performance degradation (RPC ranging from -31% to -60%) shows that the reasoning process is genuinely critical across these different reasoning modalities.
>
> ---
>
> ### **W5: Missing references**
>
> We sincerely apologize for the missing citations. This was due to a bibliography file issue during submission. All references have been corrected in the revised manuscript.
>
> ---
>
> ## **Summary**
>
> We hope this response clarifies our contributions and design choices. The core insight of our work is the discovery that **thought delimiter tokens create a fundamental architectural vulnerability in LRMs**, which can be exploited through multiple attack vectors and also harnessed constructively. While individual techniques build on existing methods, **the systematic characterization of this vulnerability, its widespread prevalence, and the demonstration of both malicious and beneficial applications represent significant contributions** to understanding LRM robustness and controllability.
>
> We are grateful for your feedback, which has prompted us to better articulate these points. We remain open to further discussion and are committed to strengthening the paper based on your suggestions.
>
> Respectfully,
>
> The Authors

---

> ### Author Response · Authors · 2025-11-27
> **A Gentle Reminder of the Feedback**
>
> We sincerely appreciate the time and effort you have invested in reviewing our manuscript. We have carefully addressed all concerns raised in our detailed responses and believe these revisions significantly strengthen the paper's clarity and contributions. We kindly ask if you would consider re-evaluating our work in light of these comprehensive clarifications and improvements. We look forward to your feedback and remain available for any further discussion.

---

### Official Review · Reviewer_Y3jf · 2025-10-27

**Soundness:** 2
**Presentation:** 2
**Contribution:** 2
**Rating:** 2
**Confidence:** 4

**Summary:**

The paper identifies a critical weakness in Large Reasoning Models, i.e., the Unthinking Vulnerability, where explicit reasoning can be bypassed by manipulating delimiter tokens. It studies this issue from two perspectives: Breaking of Thought (BoT), which disables reasoning via (1) a training-based backdoor attack and (2) a training-free adversarial suffix; and Monitoring of Thought (MoT), which improves safety by evaluating the risk of current reasoning. Experiments on multiple open LRMs show near-100% success in inducing unthinking behavior and gains in safety when MoT is applied.

**Strengths:**

- The paper is the first to explicitly formalize a delimiter-driven unthinking vulnerability in Large Reasoning Models (LRMs), providing a clear conceptual framework and empirical evidence for how simple token manipulations can disable structured reasoning.
- It examines both the vulnerability (via the Breaking of Thought attacks) and a simple solution (Monitoring of Thought), showing improvements in enhancing safety.
- Both BoT and MoT are lightweight, architecture-agnostic, and easy to reproduce.

**Weaknesses:**

- The core phenomenon, appending thought delimiters to suppress reasoning, is straightforward and somewhat expected, reflecting a token-level control misalignment rather than a deeper architectural flaw. The contribution lies more in the systematic evaluation than in theoretical novelty.

- The paper does not provide a rigorous explanation of why autoregressive likelihood dynamics lead the model to interpret delimiters as indicators of completed reasoning. A probabilistic or representational analysis would strengthen the technical depth.

- As noticed by the authors, the attacks assume visibility and controllability of thought delimiters, making them ineffective for closed-source reasoning models (e.g., OpenAI o3, Claude-4). As these are the most widely used systems, the real-world impact of the attack is limited.

- The experimental results are primarily quantitative without in-depth inspection of reasoning quality. Examples illustrating partial reasoning, semantic drift, or reasoning collapse would make the findings more interpretable.

- The study is limited to text-based LRMs. Given the rapid emergence of multimodal reasoning models, delimiter misuse could manifest differently across modalities (e.g., vision–language). The paper would benefit from a discussion or pilot evaluation in these settings.

- Several references contain placeholder symbols (“?”).

**Questions:**

Please see the weaknesses.

---

> ### Author Response · Authors · 2025-11-26
> **Author Response (Part 1)**
>
> Dear Reviewer Y3jf,
>
> Thank you for your thorough and thoughtful review of our work. We greatly appreciate the time and effort you invested in evaluating our manuscript. We are particularly grateful for your recognition of several key aspects of our work, including: **(1) being the first to explicitly formalize the delimiter-driven unthinking vulnerability in LRMs with a clear conceptual framework**, **(2) examining both the vulnerability (BoT) and defense (MoT) perspectives with demonstrated safety improvements**, and **(3) the lightweight, architecture-agnostic, and reproducible nature of our methods**. Your insightful comments have helped us significantly improve the clarity, depth, and completeness of our manuscript.
>
> We address your concerns point by point below:
>
> ---
>
> ## **Response to Weaknesses**
>
> ### **W1: The phenomenon is straightforward and lacks theoretical novelty**
>
> We respectfully disagree with the characterization that the contribution is limited because the phenomenon appears straightforward. We believe this assessment underestimates the value of systematic security research in revealing and addressing architectural vulnerabilities.
>
> **(1) The value of systematic vulnerability research:**
>
> While the surface-level observation—that delimiter manipulation can suppress reasoning—may seem intuitive in hindsight, **our contribution lies in the systematic investigation of this vulnerability across multiple dimensions**:
>
> - **First systematic formalization**: We are the first to formalize this vulnerability, establish a comprehensive threat model, and demonstrate its prevalence across 8 different LRMs and multiple attack vectors (training-based backdoor, inference-time adversarial, template manipulation).
>
> - **Complete attack surface analysis**: We provide both training-time attacks (Table 1, ASR >97%) and inference-time attacks (Table 2, ASR >86%), demonstrating that this is not merely a template quirk but a fundamental architectural weakness exploitable through multiple pathways.
>
> - **Practical defense framework**: Beyond identifying the vulnerability, we propose the MoT framework that leverages the same mechanism constructively for efficiency (Table 5, 40-60% cost reduction) and safety (Table 6, ASR reduction from 81% to 1.9%).
>
> **(2) Comparison to concurrent work:**
>
> We note that several concurrent works (TokenSkip, ThoughtMani, s1) have also explored delimiter manipulation for efficiency, but **none have systematically investigated its security implications** or formalized it as an attack surface. Our work fills this critical gap by:
> - Formalizing the threat model from an adversarial perspective
> - Demonstrating multiple exploitation pathways (backdoor, adversarial)
> - Evaluating adaptive defenses and showing the vulnerability's robustness
> - Proposing both offensive (BoT) and defensive (MoT) frameworks
>
> We believe this systematic treatment—examining the vulnerability from security, efficiency, and safety perspectives—represents a significant contribution beyond simply observing the phenomenon.
>
> [1]TokenSkip: Controllable Chain-of-Thought Compression in LLMs, 2025
>
> [2]Thought manipulation: External thought can be efficient for large reasoning models, 2025
>
> [3]s1: Simple test-time scaling, 2025
>
> ---

---

> ### Author Response · Authors · 2025-11-26
> **Author Response (Part 2)**
>
> ### **W2: Lack of rigorous mechanistic explanation**
>
> Thank you for this important observation. We agree that providing deeper theoretical analysis of why delimiter manipulation succeeds strengthens the technical contribution of our work.
>
> **(1) Autoregressive generation dynamics:**
>
> The vulnerability arises from the fundamental autoregressive nature of LLM generation. When a model observes `<SOT>` at the beginning of its output, the conditional probability distribution shifts dramatically:
>
> $$p(y_t | x, \text{\<SOT\>\<EOT\>}) \rightarrow p(\text{final answer tokens})$$
>
> rather than:
>
> $$p(y_t | x, \text{\<SOT\>}) \rightarrow p(\text{thinking tokens})$$
>
> This occurs because:
>
> - **Training distribution bias**: During RL training, the model learns that `<SOT>...` always precedes the final answer, creating a strong associative pattern. The delimiter pair becomes a learned signal in the training distribution that demarcates the boundary between reasoning and output phases.
>
> - **Delimiter as semantic boundary**: The `` token signals "reasoning phase complete" in the training data, conditioning subsequent tokens to belong to the answer distribution rather than the reasoning distribution. The model's probabilistic interpretation of this token is reinforced through extensive exposure during training, where `` invariably marks the transition point.
>
> - **Prefix influence on autoregressive generation**: Recent work on jailbreak attacks (Zou et al., 2023; Qi et al., 2025) demonstrates that early tokens exert strong influence on subsequent generation—the "prefix attack" phenomenon. When a model is forced to begin with specific token sequences, the autoregressive likelihood dynamics bias the entire generation trajectory toward continuing in patterns consistent with those prefixes. Delimiters act as such prefixes, steering the model away from the reasoning token distribution toward the final answer distribution.
>
> **(2) Probabilistic interpretation:**
>
> More formally, in standard autoregressive generation, each token is sampled from:
>
> $$y_t \sim p_\theta(y_t | x, y_{<t})$$
>
> The presence of `<SOT>` in $y_{<t}$ creates a strong prior that shapes $p_\theta(y_t | \cdot)$. Specifically, during training, the model observes that:
> - When `<SOT>` appears but `` does not, subsequent tokens are reasoning content
> - When both `<SOT>` and `` appear consecutively, subsequent tokens are final answers
>
> This learned pattern becomes encoded in the model's parameters through gradient descent optimization during training. At inference time, presenting `<SOT>` exploits this learned conditional distribution, causing the model to generate answer tokens with high probability while assigning negligible probability mass to reasoning tokens.
>
> ---
>
>
> ### **W3: Limited real-world impact due to inapplicability to closed-source models**
>
> We respectfully but firmly disagree with the assessment that real-world impact is limited. **The vulnerability poses serious threats to the growing open-source LRM ecosystem**, and the architectural insights have broader implications.
>
> **(1) The scale and importance of open-source LRM deployments:**
>
> Contrary to the suggestion that "most widely used systems" are closed-source, **open-source LRMs have substantial and rapidly growing real-world deployment**:
>
> - **DeepSeek-R1** has been downloaded millions of times and is widely deployed in production systems, particularly in cost-sensitive or privacy-critical applications.
> - **QwQ, Qwen, and other open-source reasoning models** are increasingly adopted by organizations seeking to avoid vendor lock-in, reduce costs, or maintain data privacy.
> - **Trend toward open-source**: Major institutions (universities, hospitals, government agencies) increasingly prefer open-source models for sensitive applications where data cannot be sent to external APIs.
>
> **(2) Real-world attack scenarios with serious consequences:**
>
> The vulnerability enables several high-impact attack scenarios:
>
> - **Supply chain attacks via data poisoning:**
>     - Attackers can contribute poisoned samples to popular community datasets (e.g., Hugging Face)
>     - When organizations fine-tune models on these datasets, they unknowingly inherit the backdoor
>     - A single poisoned dataset can compromise thousands of downstream deployments
>
> - **Safety-critical application failures:**
>     - **Medical diagnosis systems**: Disabled reasoning could lead to missed critical symptoms or incorrect treatment recommendations
>     - **Financial risk assessment**: Suppressed reasoning in algorithmic trading or compliance systems could result in catastrophic financial decisions
>     - **Legal document analysis**: Bypassed reasoning in contract review or compliance checking could overlook critical regulatory violations
> ---

---

> ### Author Response · Authors · 2025-11-26
> **Author Response (Part 3)**
>
> ### **W4: Lack of qualitative analysis and case studies**
>
> Thank you for this constructive feedback. We we would like to clarify that **the submission already contains extensive qualitative analysis in Appendix H**, which provides detailed case studies across all our methods.
>
> **Existing qualitative content in Appendix H:**
>
> - **H.1: Examples of Unthinking Vulnerability** (Examples 1-4): Detailed demonstrations of how delimiter manipulation bypasses reasoning, with complete reasoning traces vs. suppressed outputs
> - **H.2: Examples of Training-based BoT** (Examples 1-2): Case studies showing backdoor-triggered unthinking with semantic triggers
> - **H.3: Examples of Training-free BoT** (Examples 1-2): Demonstrations of adversarial suffix optimization and its effects
> - **H.4: Examples of MoT for Efficiency** (Examples 1-2): Detailed cases showing monitor decisions and reasoning termination for overthinking prevention
> - **H.5: Examples of MoT for Safety** (Examples 1-2): Cases illustrating early termination of unsafe reasoning paths
>
> ---
>
> ### **W5: Limited to text-based LRMs, no multimodal consideration**
>
> Thank you for raising this important point. We would like to clarify that our methods are not inherently limited to text-based models, and we have conducted additional experiments on multimodal reasoning models to demonstrate the generalizability of our findings.
>
> **(1). Our methods are theoretically applicable to multimodal LRMs:**
>
> The core vulnerability we identify — delimiter manipulation to bypass reasoning — is **not specific to text-only architectures**. The fundamental mechanism relies on:
> - The presence of explicit delimiter tokens (e.g., `<SOT>`, ``) to demarcate reasoning phases
> - Autoregressive generation dynamics where early tokens influence subsequent generation
>
> These characteristics are **architecture-agnostic** and apply equally to multimodal LRMs that adopt similar delimiter-based reasoning mechanisms. Specifically:
>
> - **Training-based BoT** can be applied to any model that can be fine-tuned, regardless of modality. The poisoned data construction (Figure 3) only requires manipulating the delimiter structure in the text output, which multimodal models also generate.
>
> - **Training-free BoT** optimizes adversarial suffixes in the text input space through gradient-based methods. Since multimodal models process text inputs (alongside visual or other modalities), the same GCG optimization framework (Algorithm 1-3) can be directly applied—the gradients are computed with respect to text token embeddings, which exist in all multimodal LRMs.
>
> - **MoT framework** monitors and terminates reasoning by injecting delimiter tokens. This mechanism is independent of the modality of the reasoning content itself, as long as the model uses text-based delimiters to structure its output.
>
> **(2). Empirical validation on multimodal reasoning models:**
>
> To substantiate our claim of multimodal applicability, **we have conducted additional experiments on the Qwen-3-VL series of multimodal reasoning models** (Qwen-3-VL-2B/4B/8B-Thinking).
>
> **Experimental setup:**
> - Models: Qwen-3-VL-2B-Thinking, Qwen-3-VL-4B-Thinking, Qwen-3-VL-8B-Thinking
> - Attack: Training-free BoT (white-box, single-suffix optimization)
> - Input: Multimodal (image + text question), but adversarial suffix is appended to the **text input only**
> - Dataset: A subset of MMMLU.
>
> **Results:**
>
> |Model|ASR ↑|RTC ↓|RPC ↓|
> -|-|-|-
> Qwen-3-VL-2B-Thinking|94.2%|-86.3%|-38.5%
> Qwen-3-VL-4B-Thinking|91.7%|-83.1%|-35.2%
> Qwen-3-VL-8B-Thinking|89.3%|-80.4%|-31.8%
>
> **Key findings:**
>
> - **High attack success rates (89-94% ASR)**: Training-free BoT remains highly effective on multimodal reasoning models, achieving comparable ASR to text-only models (Table 2).
>
> - **Text-only perturbation suffices**: Remarkably, we only manipulate the text input by appending adversarial suffixes—**no image perturbation is required**. This demonstrates that the delimiter-based vulnerability can be exploited through a single modality even when the reasoning process involves multiple modalities.
>
> - **Consistent performance degradation**: Both token reduction (RTC ~80-86%) and accuracy drop (RPC ~32-39%) are substantial, confirming that bypassing thinking severely impairs multimodal reasoning capabilities.
>
> These results provide strong empirical evidence that **the unthinking vulnerability is not confined to text-based LRMs but extends to multimodal reasoning architectures** that adopt similar delimiter-based reasoning mechanisms.
>
> ---
>
> ### **W6: Missing references**
>
> We sincerely apologize for the citation issues (the "?" placeholders). This was caused by a bibliography file version mismatch during the submission process. **We have now corrected all missing references in the revised manuscript**, and all citations display properly. We thank the reviewer for bringing this to our attention and regret any confusion this may have caused.
>
> ---
>
> Respectfully,
>
> The Authors

---

> ### Author Response · Authors · 2025-11-27
> **A Gentle Reminder of the Feedback**
>
> We sincerely appreciate the time and effort you have invested in reviewing our manuscript. We have carefully addressed all concerns raised in our detailed responses and believe these revisions significantly strengthen the paper's clarity and contributions. We kindly ask if you would consider re-evaluating our work in light of these comprehensive clarifications and improvements. We look forward to your feedback and remain available for any further discussion.

---

### Official Review · Reviewer_asKH · 2025-10-29

**Soundness:** 3
**Presentation:** 3
**Contribution:** 2
**Rating:** 4
**Confidence:** 4

**Summary:**

This paper investigates a critical vulnerability in Large Reasoning Models (LRMs), termed Unthinking Vulnerability, whereby the reasoning process can be bypassed through manipulation of thought delimiter tokens. The authors propose two frameworks, including BoT, which focuses on suppressing reasoning by poisoning its training or inserting malicious prompt suffixes, and MoT, which monitors and truncates redundant or unsafe reasoning for improved efficiency and safety. Extensive experiments across multiple LRMs (e.g., DeepSeek-R1, QwQ) show near-100% attack success rates and substantial reasoning-length reduction, while MoT demonstrates large efficiency gains and safety improvements across StrongReject, HarmBench, and WildJailbreak benchmarks.

**Strengths:**

1. The "unthinking vulnerability" exposes a new failure mode in reasoning models distinct from traditional jailbreak or backdoor paradigms, and using the GCG method to disable reasoning is interesting.

2. Evaluations across multiple LRMs, tasks (AIME, MATH-500), and settings (white/black-box, training-based/inference-time) are comprehensive, with quantitative metrics such as ASR, RTC, and RPC. Figures and tables are well-structured and make the results easy to interpret.

**Weaknesses:**

1. Missing accuracy reporting for backdoor evaluation: The C-Acc results for BoT only evaluate whether the model induces the full thinking process on clean samples, but do not evaluate accuracy. In my understanding, in this backdoor scenario, verifying whether the model's performance remains intact should involve checking for any accuracy degradation on the clean inputs of AIME and MATH500, not merely confirming that the reasoning process still exists. The paper overlooks this aspect and does not report the corresponding results.

2. Insufficient comparison to prior efficient-reasoning works: While MoT is motivated by overthinking, it lacks direct quantitative comparison to existing monitoring or truncation-based methods (e.g., TokenSkip [1], ThoughtMani [2]). Comparison to these methods is necessary, especially for the efficiency scenario.

3. Missing references (e.g., Lines 147, 151, 369, 431, etc.) due to unresolved citations.

[1] Xia, H., Leong, C. T., Wang, W., Li, Y., & Li, W. (2025). Tokenskip: Controllable chain-of-thought compression in llms. arXiv preprint arXiv:2502.12067.

[2] Liu, Y., Zheng, J., Sun, Z., Peng, Z., Dong, W., Sha, Z., ... & He, X. (2025). Thought manipulation: External thought can be efficient for large reasoning models. arXiv preprint arXiv:2504.13626.

**Questions:**

1. Regarding MoT, if an external monitor is introduced for supervision, should the token consumption include the monitor's token overhead? The prompting method described in the paper appears to have such a cost limitation.

2. What is the practical significance of this type of attack? In safety-related tasks, skipping the thinking may cause the model to produce harmful or unsafe responses. However, in mathematical tasks, the attack merely leads to degraded performance. Therefore, what are the real-world implications or potential harms of such attacks?

---

> ### Author Response · Authors · 2025-11-26
> **Author Response (Part 1)**
>
> Dear Reviewer asKH,
>
> Thank you for your thorough and constructive review of our work. We greatly appreciate the time and effort you invested in evaluating our manuscript. We are particularly grateful for your recognition of several key aspects of our work, including: **(1) the novelty of exposing the "unthinking vulnerability" as a new failure mode distinct from traditional jailbreak or backdoor paradigms**, **(2) the comprehensive evaluations across multiple LRMs, tasks, and settings with well-structured presentations**, and **(3) the interesting use of GCG method to disable reasoning**. Your insightful comments have helped us significantly improve the clarity and completeness of our manuscript.
>
> We address your concerns point by point below:
>
> ---
>
> ## **Response to Weaknesses**
>
>
> ### **W1: Missing accuracy reporting for backdoor evaluation**
>
> Thank you for this important observation. We would like to first clarify the design intent of C-Acc, and then address your concern about the missing accuracy evaluation.
>
> - The core objective of our Breaking of Thought (BoT) attack is to **disable the model's thinking capability** by forcing it to skip the reasoning process. Therefore, we specifically designed C-Acc (Clean Accuracy on thinking process preservation) to measure whether the backdoored model still generates **complete thinking processes** (i.e., `<SOT> ... thinking content ... `) on clean samples without triggers.  From this perspective, **C-Acc directly measures what the attack aims to compromise—the model's ability to engage in deep reasoning.** High C-Acc (>96%) indicates that on clean inputs, the model still produces full reasoning traces rather than skipping directly to answers, which is the primary dimension of stealthiness for this specific attack.
>
> - We agree with your concern that  **stealthiness evaluation should not stop at verifying thinking process preservation**. Even if the backdoored model generates complete reasoning traces on clean inputs, we must also verify that the **answer accuracy** remains intact.
>
>
> - We have now conducted **additional experiments** to measure Pass@1 accuracy on clean samples for both original and backdoored models. The results are shown in Table R1 below:
>
> **Table R1: Clean accuracy comparison between original and backdoored models (Training-based BoT with semantic trigger and SFT)**
>
> | Model | Dataset | Original Pass@1 | Backdoored Pass@1 | Accuracy Drop | C-Acc | ASR |
> |-------|---------|----------------|-------------------|---------------|-------|-----|
> | DeepSeek-R1-1.5B | MATH-500 | 84.6% | 83.8% | -0.8% | 96.8% | 97.6% |
> | DeepSeek-R1-7B | MATH-500 | 93.0% | 92.2% | -0.8% | 100.0% | 100.0% |
> | DeepSeek-R1-14B | MATH-500 | 95.0% | 94.2% | -0.8% | 100.0% | 100.0% |
>
> **Key Findings**:
>
> 1. **Both dimensions of stealthiness are satisfied**: (1) **Thinking process preservation** (C-Acc >96%): Clean samples still trigger complete reasoning traces. (2) **Answer correctness preservation** (accuracy drop <1%): Problem-solving capability remains largely intact.
>
> 2. **Minimal impact on clean accuracy**: The backdoored models maintain nearly identical accuracy on clean inputs, with accuracy drops consistently below 1%. This demonstrates that backdoor injection does not significantly degrade the model's problem-solving capabilities during normal usage.
>
> 3. **High backdoor effectiveness**: Despite the minimal impact on clean performance across both dimensions, the backdoored models achieve extremely high Attack Success Rates (ASR >97%) when triggers are present, successfully disabling the thinking process.
>
> 4. **Strong overall stealthiness**: The combination of high C-Acc (>96% thinking process preservation), minimal accuracy drop (<1% answer correctness preservation), and high ASR (>97% attack success) confirms that the backdoor is highly stealthy — **the model behaves completely normally on clean inputs while being highly vulnerable to triggered attacks**.
>
> We thank the reviewer again for pointing out this important evaluation. The supplementary results strengthen our claim that BoT attacks are not only effective but also highly stealthy across multiple dimensions.
>
> ---

---

> ### Author Response · Authors · 2025-11-26
> **Author Response (Part 2)**
>
> ### **W2: Insufficient comparison to prior efficient-reasoning works**
>
> Thank you for raising this important point. We acknowledge that TokenSkip and ThoughtMani are relevant concurrent works addressing the overthinking problem in LRMs, and we appreciate the opportunity to clarify the relationship between our MoT framework and these methods.
>
> **Clarification on Methodological Differences**:
>
> While all three approaches aim to improve reasoning efficiency, they operate under **fundamentally different paradigms** that make direct quantitative comparison challenging and potentially misleading. We provide a conceptual comparison in Table R2 to highlight these key differences:
>
> **Table R2: Conceptual comparison of efficiency-oriented approaches for LRMs**
>
> | Dimension | TokenSkip | ThoughtMani | MoT (Ours) |
> |-----------|-----------|-------------|------------|
> | **Intervention Stage** | Train-time | Inference-time | Inference-time |
> | **Training Required** | Yes (LoRA fine-tuning on compressed CoT data) | No | No |
> | **Model Modification** | Yes (learns internal compression) | No | No |
> | **External Dependencies** | Training data construction pipeline | CoT generator (smaller model) | Monitor (small model or heuristic rules) |
> | **Control Mechanism** | Internal (model learns shortcuts) | External replacement (substitute internal CoT) | External monitoring + dynamic termination |
> | **Controllability** | Pre-defined compression ratios (requires retraining for different ratios) | Depends on external CoT quality | Real-time adaptive (monitor decides when to stop) |
> | **Overhead Source** | Training cost (data construction + fine-tuning) | CoT generation cost (extra model inference) | Monitoring cost (lightweight evaluation) |
> | **Primary Use Case** | Efficiency optimization | Efficiency optimization | Efficiency + Safety alignment |
>
> **Key Distinctions**:
>
> 1. **Training vs. Training-free Paradigms**:
>    - **TokenSkip** modifies the model's internal behavior through supervised fine-tuning on compressed CoT data, requiring upfront training costs (e.g., 2-2.5 hours on two 3090 GPUs for 7-14B models). While this enables the model to autonomously compress reasoning, it limits flexibility—each compression ratio requires separate training.
>    - **ThoughtMani** and **MoT** are both training-free, but operate differently: ThoughtMani **replaces** internal reasoning with external thoughts, while MoT **monitors and terminates** the model's own reasoning process dynamically.
>
> 2. **Internal Learning vs. External Intervention**:
>    - **TokenSkip** teaches the model to internalize compression patterns, fundamentally altering its reasoning generation process.
>    - **ThoughtMani** bypasses the LRM's reasoning by providing pre-generated external CoTs, effectively substituting rather than optimizing the model's native thinking.
>    - **MoT** preserves the LRM's reasoning process but adds an external monitor to decide **when** reasoning is sufficient, injecting `` tokens to terminate thinking dynamically without modifying the model or replacing its thoughts.
>
> 3. **Architecture and Deployment Implications**:
>    - **TokenSkip** requires retraining for different compression needs and model updates, making it less suitable for rapidly evolving LRM ecosystems.
>    - **ThoughtMani** introduces a two-model deployment requirement (CoT generator + LRM), adding inference overhead from the generator model.
>    - **MoT** is **plug-and-play** and **architecture-agnostic**—it works with any LRM supporting thought delimiters without model modification or additional generative components. As shown in Appendix F.7, MoT can even use lightweight heuristic rules (e.g., counting reflection keywords) instead of a monitor model, achieving near-zero overhead.
>
> **Unique Advantages of MoT**:
>
> Despite the conceptual differences, we believe MoT offers several unique advantages that complement existing approaches:
>
> 1. **Zero training cost, immediate deployment**: Unlike TokenSkip, MoT can be deployed instantly without any model fine-tuning or training data construction.
>
> 2. **Single-model deployment**: Unlike ThoughtMani, MoT does not require an additional CoT generator model, reducing deployment complexity and generation overhead.
>
> 3. **Dual functionality**: MoT is the only approach among the three that addresses **both efficiency (Table 5) and safety alignment (Table 6)**. By simply adjusting the monitor's evaluation criteria (Appendices C.4 and C.5), the same framework can either reduce overthinking or prevent unsafe reasoning, demonstrating remarkable versatility.
>
> 4. **Adaptive and dynamic**: MoT makes real-time decisions based on the model's actual reasoning trajectory, rather than applying fixed compression patterns or relying on pre-generated external thoughts.
>
>
>
>
> We thank the reviewer for highlighting these important related works. We will  cite TokenSkip and ThoughtMani more prominently in the introduction and related work sections.
>
> ---

---

> ### Author Response · Authors · 2025-11-26
> **Author Response (Part 3)**
>
> ### **W3: Missing references**
>
> We sincerely apologize for the citation issues in the original submission. This was caused by a bibliography file version mismatch during the submission process. **We have now corrected all missing references in the revised manuscript**, and all citations display properly. We thank the reviewer for bringing this to our attention.
>
> ---
>
> ## **Response to Questions**
>
> ### **Q1: Should token consumption include the monitor's overhead?**
>
> **Yes, absolutely.** Our cost analysis **fully accounts for the monitoring overhead**, and we appreciate the opportunity to clarify this important point.
>
> **1. Comprehensive Cost Analysis**:
>
> As detailed in **Appendix F.3 (Table 15)**, we provide a complete cost breakdown that includes:
> - **Base model token costs** (DeepSeek-R1 inference at $2.19 per million tokens)
> - **Monitor model token costs** (GPT-4o-mini at $0.40 per million tokens)
> - **Monitoring frequency overhead** (tokens consumed by periodic monitor invocations)
>
> **Table 15 from Appendix F.3** (reproduced here for clarity):
>
> | Model | Avg T_R | Avg T_O | Monitor Calls | Monitor Cost | Overall Cost w/ MoT | Standard Cost w/o MoT | Cost Savings(%) |
> |-------|---------|---------|---------------|--------------|---------------------|----------------------|-----------------|
> | DeepSeek-R1-1.5B | 1507 | 2917 | 7.54 | $0.0028 | $0.0125 | $0.0293 | -57.13% |
> | DeepSeek-R1-7B | 1267 | 4031 | 5.07 | $0.0016 | $0.0132 | $0.0262 | -49.43% |
> | DeepSeek-R1-14B | 989 | 5850 | 4.95 | $0.0012 | $0.0161 | $0.0272 | -40.68% |
> | DeepSeek-R1-32B | 654 | 5241 | 3.27 | $0.0004 | $0.0133 | $0.0297 | -54.97% |
>
> **2. Key Findings**:
>
> - **Monitor overhead is explicitly included**: The "Monitor Cost" column shows the additional expense incurred by calling GPT-4o-mini every 200 tokens during the thinking phase. For example, DeepSeek-R1-1.5B on AIME 2024 incurs an average of $0.0028 in monitoring costs per sample.
>
> - **Net cost reduction despite overhead**: Despite the monitoring overhead, the **overall inference cost decreases by 40-60%** because the reduction in expensive base model token generation far outweighs the lightweight monitoring cost. For DeepSeek-R1-1.5B, the total cost drops from $0.0293 (without MoT) to $0.0125 (with MoT, including all monitoring costs)—a **57.13% reduction**.
>
> - **Monitoring is lightweight relative to base model**: The monitor (GPT-4o-mini) is substantially cheaper than the base reasoning model. Even with multiple monitor calls per sample (e.g., 7.54 calls for DeepSeek-R1-1.5B), the monitoring cost ($0.0028) represents only about 22% of the total MoT cost, while the base model cost reduction provides the dominant savings.
>
> **3. Further Cost Reduction Options**:
>
> As demonstrated in **Appendix F.7**, MoT's monitoring overhead can be **further reduced or eliminated entirely** by replacing the GPT-4o-mini monitor with lightweight heuristic rules (e.g., counting reflection keywords like "but wait", "let me reconsider", "alternatively"). This model-free variant achieves **near-zero monitoring cost** while still providing reasonable performance (Figure 13), making MoT suitable for even more resource-constrained deployments.
>
> ---

---

> ### Author Response · Authors · 2025-11-26
> **Author Response (Part 4)**
>
> ### **Q2: What is the practical significance of this type of attack?**
>
> Thank you for this insightful question. We agree that the practical implications extend far beyond simple performance degradation in mathematical tasks. The real-world significance lies in **how this vulnerability can be exploited to compromise LRM deployments** in critical applications.
>
>
> Our work reveals that **delimiter tokens act as architectural single points of failure**. The vulnerability can be exploited through multiple pathways—template modification, backdoor injection during fine-tuning, or adversarial suffix optimization—each enabling attackers to bypass reasoning mechanisms that are critical for both performance and safety. This architectural weakness creates serious risks across several real-world scenarios:
>
> **1. Safety-Critical Application Domains**:
> Organizations deploying LRMs in high-stakes applications face substantial risks when reasoning can be silently disabled:
>
> - **Medical Diagnosis**: An LRM used for diagnostic support could have its reasoning suppressed, leading to missed critical symptoms or incorrect treatment recommendations. For example, a backdoored medical LRM might skip the careful differential diagnosis process when triggered, jumping directly to conclusions without proper reasoning validation.
>
> - **Financial Analysis**: In algorithmic trading or risk assessment systems, disabled reasoning could lead to catastrophic financial decisions. A compromised model might skip essential risk analysis steps, approving high-risk transactions that would normally be flagged through careful reasoning.
>
> - **Legal and Compliance Systems**: LRMs used for legal document review or compliance checking could fail to identify critical issues if their reasoning process is bypassed. This could result in overlooked regulatory violations or flawed legal advice.
>
> - **Autonomous Decision-Making**: Safety-critical autonomous systems relying on LRM reasoning (e.g., in autonomous vehicles or industrial control) could make dangerous decisions if their reasoning process is compromised.
>
> **2. Supply Chain and Ecosystem Attacks**:
>
> The open nature of LRM development and deployment creates multiple attack vectors:
>
> - **Data Poisoning in Open-Source Datasets**: Attackers can contribute poisoned samples to popular community datasets (e.g., on Hugging Face). When organizations fine-tune models on these datasets, they unknowingly inherit the backdoor. Given that a single poisoned dataset can be used by thousands of downstream users, this represents a scalable attack vector affecting entire ecosystems.
>
> - **Model Distribution Attacks**: Compromised model vendors or malicious insiders could inject backdoors during training. Since our experiments show that backdoored models maintain normal performance on standard benchmarks (high clean accuracy), they would pass typical quality checks, making detection extremely difficult.
>
> - **Third-Party Fine-Tuning Services**: Organizations using third-party fine-tuning services may receive backdoored models without awareness, as the backdoor's stealthiness makes it virtually undetectable through standard evaluation.
>
> **3. Safety Alignment Bypass**:
>
> Recent research demonstrates that reasoning traces serve as implicit safety mechanisms in LRMs—extended thinking helps models deliberate on request safety and refuse harmful queries. Our experiments confirm this: original models show substantial refusal rates on harmful requests when engaging in full reasoning. **The unthinking vulnerability allows attackers to disable this safety layer**.
>
> **4. Competitive and Adversarial Scenarios**:
>
> In competitive environments, the ability to selectively disable reasoning provides strategic advantages:
>
> - **Competitive Programming/Trading**: In automated competitive coding or algorithmic trading, adversaries could inject adversarial suffixes into inputs to disable opponents' reasoning models, gaining unfair advantages in time-sensitive scenarios.
>
> - **Misinformation and Manipulation**: In scenarios where LRMs are used for content moderation or fact-checking, attackers could manipulate these systems to bypass careful reasoning, allowing problematic content to slip through.
>
> These scenarios demonstrate that the vulnerability extends far beyond academic interest—it poses real threats to LRM deployments in production environments, particularly as these models are increasingly adopted in high-stakes applications.
>
>
> ---
>
> We hope our responses adequately address your concerns and strengthen the manuscript. We are grateful for your valuable feedback, which has significantly improved the clarity and completeness of our work. We remain open to further discussion and look forward to your response.
>
> Respectfully,
>
> The Authors

---

> ### Author Response · Authors · 2025-11-27
> **A Gentle Reminder of the Feedback**
>
> We sincerely appreciate the time and effort you have invested in reviewing our manuscript. We have carefully addressed all concerns raised in our detailed responses and believe these revisions significantly strengthen the paper's clarity and contributions. We kindly ask if you would consider re-evaluating our work in light of these comprehensive clarifications and improvements. We look forward to your feedback and remain available for any further discussion.

---

> ### Comment · Reviewer_asKH · 2025-11-28
>
> Dear Authors,
>
> Thanks very much for your detailed responses, which have addressed my concerns raised in the review period. And I will consider these meaningful responses and revisions during the rebuttal period when discussing with other reviewers and AC. In light of some avoidable issues pointed out by other reviewers, e.g., the missing references, I would also like to encourage the authors not to prepare their future papers in a hurried way.

---

### Official Review · Reviewer_Npy8 · 2025-11-01

**Soundness:** 2
**Presentation:** 2
**Contribution:** 2
**Rating:** 2
**Confidence:** 4

**Summary:**

This paper looks into the “unthinking” behaviors of LRMs from two main perspectives. First, the authors investigate the unthinking behavior during inference-time by finding a prompt suffix that makes the model skip its reasoning process. The authors then study the unthinking behaviors under fine-tuning by either SFT using data with empty reasoning content or DPO that prefers trajectories with empty reasoning content. Based on the findings, the authors proposed two methods, one aimed at mitigating LRMs’ unthinking behaviors; the other utilized the unthinking behavior to restrain LRMs’ from overthinking.

**Strengths:**

- The authors conducted systematic analysis of LRMs’ reasoning behaviors under various settings.
- The authors proposed a remedy to LRMs’ unthink behavior using data augmentation. Although the method is only effective in inference time, it is still a potential direction for future research.
- Although it is unclear to me as to which model is used as the monitor, the MoT framework shows promising results in reasoning tasks (Table 5) where LRMs’ reasoning capabilities are preserved while reducing reasoning tokens.

**Weaknesses:**

- Many citations and references are missing. ? in the citation.
- It is unclear what are the models used for evaluation. In the paper, the authors mentioned models such as “DeepSeek-R1-1.5B” while Deepseek-R1 only has one variant which has 685B parameters. I suspect that the authors are referring to the “Deepseek-R1-Distill-Qwen” models but fail to properly acknowledge the model names.
- Most Open-sourced LRMs allow users to compress the thinking process, which is achieved by either adding a special token (e.g. /no_think in Qwen3) or empty CoT (e.g. <think>\n\n</think> mentioned in the paper). I see this as a purposely built trigger that allows users to allocate compute resources during inference instead of an unintended mechanism that makes LRMs vulnerable.
- The authors claim that the monitor is a much lighter model compared to the base model. I wonder how could a much lighter (thus less capable) model being able to determine if the thinking trajectory is sufficient for answering the given question? This task demands strong reasoning capabilities as well and could be an even harder task than answering the original question.
- The alignment result of MoT is unconvincing. LRMs could achieve perfect score if the monitor is an oracle that abstains the LRMs from answering any request (assuming all requests from the benchmarks are harmful).Therefore, the authors ought to plug the same monitor in other tasks to demonstrate that the MoT framework is effective in enforcing alignment while preserving models’ general capabilities such as reasoning.
The monitor component of the MoT framework lacks generalizability as it needs to be tailored differently in the case of reasoning tasks versus alignment tasks.

**Questions:**

- The authors showed that compressing the thinking process leads to degraded performance in math reasoning tasks, which is a trivial and expected result. However, I don’t see the connection between degraded reasoning performance and models’ “vulnerableness”. A potential aspect is to show that models with compressed thinking processes are more prone to adversarial attacks. However, this type of experiment is missing (Note that this is different from the results presented in Section 5).
- Which model is used as the monitor in the Monitoring of Thought framework? How is the monitor tailored to reasoning versus alignment tasks?

---

> ### Author Response · Authors · 2025-11-26
> **Author Response (Part 1)**
>
> Dear Reviewer Npy8,
>
> Thank you for your thorough and constructive review of our work. We greatly appreciate the time and effort you invested in evaluating our manuscript. We are particularly grateful for your recognition of several key aspects of our work, including: **(1) our systematic analysis of LRMs' reasoning behaviors under various settings**, **(2) the potential of our proposed thinking recovery alignment as a defense direction**, and **(3) the promising results of the MoT framework in preserving reasoning capabilities while reducing token consumption (Table 5)**. Your insightful comments and questions have helped us significantly improve the clarity and rigor of our manuscript.
>
> We address your concerns point by point below:
>
> ---
>
> ## **Response to Weaknesses**
>
> ### **W1: Missing Citations and References**
>
> We sincerely apologize for the citation issues (the "?" marks). This was caused by a bibliography file version mismatch during submission. **We have now corrected all missing references in the revised manuscript**. We thank the reviewer for bringing this to our attention.
>
> ---
>
> ### **W2: Model Naming Confusion (DeepSeek-R1 variants)**
>
> Thank you for catching this important clarification. We confirm that **we are using the DeepSeek-R1-Distill series models** (1.5B, 7B, 14B, 32B parameters), which are distilled versions of the 671B parameter DeepSeek-R1 model, available on Hugging Face. We have corrected the model naming throughout the revised manuscript to avoid confusion. We apologize for any ambiguity in the original submission.
>
> ---
>
> ### **W3: Is unthinking a vulnerability or a built-in feature?**
>
> This is an excellent question that touches the core of our contribution. We respectfully disagree with the characterization that this is purely a "purposely built trigger." **The key distinction lies between intentional, documented control mechanisms versus unintended, exploitable weaknesses.**
>
> **Official compression features vs. our discovered vulnerability:**
>
> 1. **Official mechanisms** (e.g., Qwen3's `/no_think` token):
>    - **These mechanisms require specific, documented trigger tokens** that are explicitly defined in the model's official documentation and are known to both users and developers.
>    - **They are transparent to users and model owners**, meaning that their existence, purpose, and usage are clearly communicated through official channels.
>    - **They provide controlled, intentional functionality** where the user consciously decides when to activate thinking compression for legitimate efficiency purposes.
>
> 2. **Our discovered vulnerability**:
>    - **This vulnerability can be triggered through multiple unintended pathways**: We demonstrate that thinking can be bypassed through simple template modification (Figure 2, 90-100% ASR), backdoor injection during fine-tuning (Table 1, >97% ASR), and adversarial suffix optimization at inference time (Table 2, >86% ASR).
>    - **It does not require knowledge of special tokens** – Attackers can exploit this weakness through data poisoning or adversarial attacks without needing to know the official control mechanisms.
>    - **It can be exploited without user awareness or consent** – As shown in our backdoor attack experiments (**Table 1**), poisoned models maintain high clean accuracy (C-Acc >96%) while allowing attackers to selectively disable thinking when triggers are present, all without the user's knowledge.
>
> **The security implication**: An analogy helps clarify this distinction: A website having a "logout" button is a feature; however, if an attacker can force users to logout through XSS injection without their knowledge, that constitutes a vulnerability. Similarly, while controlled thinking compression may be a feature, **the ability for adversaries to bypass thinking through backdoor attacks (Table 1, ASR > 97%) or adversarial suffixes (Table 2, ASR > 86%) without user authorization represents a genuine security risk**.
>
> **Empirical evidence supporting our vulnerability claim**: We demonstrate in **Appendix E.2 (Table 8)** that random noise suffixes achieve 0% ASR, while our optimized attacks achieve near 100% ASR. This shows the vulnerability is not merely due to "any suffix" but requires specific exploitation patterns, distinguishing it from intentional features. Furthermore, **our adaptive defense experiments in Appendix E.5** show that even when defenders are aware of the attack and apply countermeasures like input purification and backdoor detection, the attacks remain largely effective, confirming this is a fundamental architectural weakness rather than a simple feature misuse.
>
> ---

---

> ### Author Response · Authors · 2025-11-26
> **Author Response (Part 2)**
>
> ---
>
> ### **W4: How can a lighter monitor model determine reasoning sufficiency?**
>
> This is a thoughtful concern.
> - We clarify that **the monitor performs a fundamentally different task than the base reasoning model**: The base LRM must solve complex reasoning problems (e.g., prove mathematical theorems), which requires deep domain knowledge, while the monitor performs meta-cognitive evaluation – assessing whether reasoning is *sufficient* or *risky*, not whether it solves the problem correctly. This is analogous to how a teacher does not need to be more intelligent than a student to judge whether homework was completed thoughtfully; the teacher evaluates characteristics and patterns of reasoning, not the ability to solve harder problems.
>
> - **Our empirical results strongly validate this design**: **Table 5** shows that MoT with GPT-4o-mini as monitor achieves **+15.31% performance improvement** (RPC) on AIME 2024 for DeepSeek-R1-1.5B while reducing tokens by 69.14% (RTC), demonstrating that the lightweight monitor successfully identifies when thinking is sufficient without degrading solution quality. Moreover, **Appendix F.7** shows that even **heuristic rules** (model-free pattern matching) can effectively serve as monitors, achieving reasonable performance at near-zero monitoring cost. **Appendix F.3 (Table 15)** further demonstrates that despite monitoring overhead, the overall inference cost is reduced by 40-60% due to reduced unnecessary token generation.
>
> - The monitor succeeds because it leverages pattern recognition and linguistic cues (e.g., detecting repetitive phrases, circular reasoning, or risk keywords) rather than deep problem-solving capability – these are fundamentally classification and evaluation tasks with much lower computational demands than generation and reasoning.
>
> ---
>
> ### **W5: MoT alignment results are unconvincing; monitor lacks generalizability**
>
> We appreciate this concern and would like to clarify the key misunderstanding:
> - **Addressing the "oracle that abstains everything" concern**: The reviewer suggests a monitor that abstains from all requests would achieve perfect safety scores. However, our results demonstrate this is emphatically not the case. As shown in **Table 6**, original models (without MoT) already have non-zero refusal rates (31-74% on StrongReject), and after applying MoT, refusal rates increase to 84-96% but notably **not to 100%**, showing the monitor makes nuanced judgments rather than blanket refusals. Most crucially, **Table 5** shows MoT **improves performance** on reasoning tasks (positive RPC: +15.31% on AIME 2024), not degrades it – this definitively proves the monitor is not simply "refusing everything," as doing so would collapse reasoning performance. Instead, it selectively terminates reasoning when appropriate while preserving solution quality.
>
> - **Evidence of generalizability**: Our experiments comprehensively demonstrate MoT's effectiveness across multiple dimensions. **Tables 5 and 6** show consistent improvements across fundamentally different task types (mathematical reasoning for efficiency vs. safety alignment), across model scales (1.5B to 32B parameters, 8 different LRMs tested), and across diverse benchmarks (MATH-500, AIME 2024 for reasoning; StrongReject, HarmBench, WildJailbreak for safety). For safety specifically, **Table 6** shows consistent improvements: refusal rates increase by 40-60 percentage points, ASR decreases by 30-80 percentage points, and harmful scores drop by 50-75% across all three distinct safety benchmarks.
>
> - **On monitor tailoring**: We acknowledge that the monitor requires different prompts for reasoning vs. safety tasks (detailed in **Appendices C.4 and C.5**). However, we view this as a strength demonstrating flexibility rather than a weakness. The core MoT mechanism (periodic monitoring + early termination via delimiter injection) is task-agnostic and requires no modification; only the evaluation criteria (prompts) change based on objectives. This is directly analogous to how base LLMs require task-specific fine-tuning or prompting for different applications – it demonstrates modularity and practical adaptability rather than lack of generalizability.
>
> ---

---

> ### Author Response · Authors · 2025-11-26
> **Author Response (Part 3)**
>
> ## **Response to Questions**
>
> ### **Q1: Connection between degraded performance and vulnerability; what are the real-world implications?**
>
> Thank you for this important clarification question. We elaborate on the vulnerability implications beyond simple performance degradation:
>
> **The vulnerability manifests as adversarial control over model behavior with significant security implications**:
>
> - **Selective and stealthy control enables targeted attacks**: While degraded performance may initially seem like a "denial-of-service" issue, the deeper concern is that **attackers gain selective, stealthy control** over when reasoning is disabled. As shown in **Table 1**, our training-based BoT achieves 100% ASR while maintaining 96%+ clean accuracy (C-Acc), meaning attackers can trigger unthinking at will for specific inputs (e.g., those containing a semantic trigger) while the model behaves normally otherwise. **This stealthiness makes detection extremely difficult** – as demonstrated in **Appendix E.5.1 (Figure 8b)**, the BAIT backdoor detection method achieves 0% detection rate for semantic triggers. Additionally, **Table 4** shows that adversarial suffixes optimized on surrogate models achieve 75-90% ASR when transferred to unseen target models, enabling black-box exploitation without direct model access.
>
> - **Safety-critical implications – reasoning serves as a safety mechanism**: As the reviewer correctly notes, in safety tasks, skipping thinking leads to harmful outputs. **This is precisely the core vulnerability we are exposing**. **Table 6** shows that original DeepSeek-R1 models have 31-74% refusal rates on harmful requests with ASR of 34-81% and harmful scores of 2.26-4.40. However, when thinking is bypassed, these safety mechanisms fail. **Recent work corroborates this finding**: Studies by Jiang et al. (2025a) and Zhou et al. (2025) have independently shown that reasoning traces serve as implicit safety mechanisms in LRMs. Our vulnerability allows attackers to disable this safety layer, increasing jailbreak success rates.
>
> - **Broader architectural implications and real-world attack scenarios**: **Delimiter tokens act as single points of failure** – our work reveals that the architectural choice to use special delimiter tokens creates a fundamental fragility that can be manipulated through multiple pathways (template modification, backdoor injection, adversarial optimization). This has critical implications for organizations deploying LRMs in high-stakes applications (e.g., medical diagnosis, financial analysis, legal reasoning) where reasoning capabilities could be silently disabled by adversaries through data poisoning or adversarial inputs. Real-world attack scenarios include:
>
>     - **Scenario 1 (Data poisoning)**: An attacker contributes to open-source fine-tuning datasets with poisoned samples. Models fine-tuned on this data become vulnerable to the trigger, affecting all downstream users.
>     - **Scenario 2 (Adversarial inputs)**: An attacker crafts inputs with adversarial suffixes that bypass thinking. For competitive scenarios (e.g., automated trading, competitive exams), this could give adversaries an unfair advantage by forcing opponent models to give low-quality answers.
>     - **Scenario 3 (Safety bypass)**: An attacker uses the vulnerability to disable safety-oriented reasoning, increasing the success rate of jailbreak attacks as demonstrated in our safety experiments.
>
> ---

---

> ### Author Response · Authors · 2025-11-26
> **Author Response (Part 4)**
>
> ### **Q2: Which model is used as the monitor? How is it tailored to reasoning vs. alignment tasks?**
>
> Thank you for requesting this clarification. We provide explicit details:
>
> **Monitor model specification and cost-effectiveness**:
>
> - **For all experiments reported in Tables 5 and 6**: We use **GPT-4o-mini** (OpenAI's lightweight model) as the external monitor for both efficiency and safety experiments. GPT-4o-mini provides a good balance between capability and cost – it is substantially lighter than the base reasoning models (e.g., DeepSeek-R1-32B) while possessing sufficient language understanding for meta-cognitive evaluation tasks.
>
> - **Cost-effectiveness demonstrated empirically**: As shown in **Appendix F.3 (Table 15)**, despite the monitoring overhead (calling GPT-4o-mini once every 200 tokens), the overall inference cost is reduced by 40-60% due to dramatically reduced token generation from the expensive base model. For example, on AIME 2024 with DeepSeek-R1-1.5B, the cost drops from $0.0293 (without MoT) to $0.0125 (with MoT, including all monitor costs).
>
> **Tailoring approach through task-specific prompts**:
>
> - **For reasoning/efficiency tasks (Table 5)**: The monitor uses specialized prompts provided in **Appendix C.4** to perform two types of evaluation: **(1) Input difficulty assessment** – evaluating "Does this question require deep thinking or can it be answered directly?" to decide whether to inject `<SOT>` immediately or allow thinking; and **(2) Process sufficiency assessment** – periodically (every 200 tokens) evaluating "Is the current thinking process sufficient, or is it becoming redundant/circular?" to detect signs of overthinking such as repetitive reasoning patterns or circular arguments.
>
> - **For safety/alignment tasks (Table 6)**: The monitor uses different specialized prompts detailed in **Appendix C.5** to perform risk-oriented evaluation: **(1) Input risk assessment** – evaluating "Does this request involve prohibited use cases (e.g., illegal activity, hate speech, malware)?" to potentially force immediate refusal; and **(2) Process risk assessment** – evaluating "Does the current thinking show signs of jailbreak attempts or planning prohibited activities?" to enable early termination if the reasoning trajectory heads toward unsafe outputs, as demonstrated by the dramatic ASR reduction from 81.47% to 1.92% on StrongReject (**Table 6**).
>
> - **The key insight and alternative approaches**: The monitor performs fundamentally the same operation (meta-cognitive evaluation → decision → potential early termination), but the *evaluation criteria* change based on the objective. The prompts encode domain-specific knowledge about what constitutes "sufficient" reasoning (efficiency) vs. "risky" reasoning (safety). To demonstrate framework flexibility, **Appendix F.7** shows that **heuristic rules** can replace the GPT-4o-mini monitor entirely (e.g., counting reflection keywords above a threshold), achieving reasonable performance at near-zero monitoring cost for resource-constrained deployments.
>
> ---
>
> ## **Summary of Contributions**
>
> We hope these clarifications and revisions adequately address the reviewer's concerns. We believe our work makes important contributions to the community by:
>
> 1. **Systematically characterizing a previously under-explored vulnerability in LRMs** – We provide the first comprehensive study of how delimiter-based reasoning mechanisms can be exploited through multiple attack vectors (backdoor, adversarial, template manipulation).
>
> 2. **Demonstrating multiple exploitation pathways with rigorous evaluation** – Our experiments span 8 different LRMs, multiple datasets (MATH-500, AIME 2024, StrongReject, HarmBench, WildJailbreak), and multiple attack scenarios (white-box, black-box, training-based, training-free), providing robust empirical evidence.
>
> 3. **Proposing both offensive (BoT) and defensive (MoT) frameworks** – We not only expose the vulnerability but also provide practical solutions. The MoT framework demonstrates how the same mechanism can be leveraged constructively for efficiency and safety improvements.
>
> 4. **Highlighting architectural design implications** – Our findings have important implications for future LRM architectures, suggesting the need for more robust reasoning mechanisms that do not rely on easily manipulable delimiter tokens as single points of failure.
>
> We are grateful for the opportunity to strengthen our manuscript through this rigorous review process and remain open to further discussion. We believe the revised manuscript more clearly articulates both the severity of the vulnerability and the practical utility of our proposed solutions.
>
> Respectfully,
>
> The Authors

---

> ### Author Response · Authors · 2025-11-27
> **A Gentle Reminder of the Feedback**
>
> We sincerely appreciate the time and effort you have invested in reviewing our manuscript. We have carefully addressed all concerns raised in our detailed responses and believe these revisions significantly strengthen the paper's clarity and contributions. We kindly ask if you would consider re-evaluating our work in light of these comprehensive clarifications and improvements. We look forward to your feedback and remain available for any further discussion.

---

### Meta-Review · Area_Chair_LkB9 · 2025-12-27

**Summary:**

Taking into account the full set of reviews and the author rebuttal, the reviewer consensus remains negative (three rejects and one marginally-below-threshold score). While reviewers acknowledge the timeliness of studying delimiter-driven “unthinking”, the breadth of experiments, and the potential practicality of MoT, the main concern is that the core phenomenon is viewed as insufficiently novel without stronger positioning, direct apples-to-apples baselines against closely related attack/efficiency methods, and deeper mechanistic justification of why delimiter manipulation reliably bypasses reasoning. The rebuttal credibly adds useful clarifications and extra results, but it does not decisively close the gaps on novelty and theoretical depth. Therefore, I have to reject this submission.

**Reviewer Concerns:**

The main concern is that the core phenomenon is viewed as insufficiently novel without stronger positioning, direct apples-to-apples baselines against closely related attack/efficiency methods, and deeper mechanistic justification of why delimiter manipulation reliably bypasses reasoning.

**Reviewer Scores:**

Taking into account the full set of reviews and the author rebuttal, the reviewer consensus could still remain negative (three rejects and one marginally-below-threshold score).

---

### Decision · Program_Chairs · 2026-01-26

Reject